# COVID-19 and human-nature relationships: Vermonters' activities in nature and associated nonmaterial values during the pandemic

Joshua W. Morse[1,2]*, Tatiana M. Gladkikh[1,2], Diana M. Hackenburg[1,2], Rachelle K. Gould[1,2,3]

1 University of Vermont, The Rubenstein School of Environment and Natural Resources, Burlington, Vermont, United States of America, 2 Gund Institute for Environment, Burlington, Vermont, United States of America, 3 Environmental Program, Bittersweet House, Burlington, Vermont, United States of America

* joshua.morse@uvm.edu

**Data Availability Statement:** All relevant data are within the manuscript and its Supporting Information files.

## Abstract

The COVID-19 pandemic has rapidly modified Earth's social-ecological systems in many ways; here we study its impacts on human-nature interactions. We conducted an online survey focused on peoples' relationships with the non-human world during the pandemic and received valid responses from 3,204 adult residents of the state of Vermont (U.S.A.). We analyzed reported changes in outdoor activities and the values associated with human-nature relationships across geographic areas and demographic characteristics. We find that participation increased on average for some activities (foraging, gardening, hiking, jogging, photography and other art, relaxing alone, walking, and watching wildlife), and decreased for others (camping, relaxing with others). The values respondents ranked as more important during the pandemic factored into two groups, which we label as "Nurture and Recreation values" and "Inspiration and Nourishment values." Using multinomial logistic regression, we found that respondents' preferences for changes in activity engagement and value factors are statistically associated with some demographic characteristics, including geography, gender, income, and employment status during the pandemic. Our results suggest that nature may play an important role in coping during times of crisis, but that the specific interactions and associated values that people perceive as most important may vary between populations. Our findings emphasize for both emergency and natural resources planning the importance of understanding variation in how and why people interact with and benefit from nature during crises.

## Introduction

The COVID-19 pandemic has, in a few months, changed our planet's social-ecological systems in substantial ways [1]. Stay-at-home orders, dramatic shifts in work and social schedules, and restrictions on the use of public spaces intertwine to modify when, where, and how people interact with the world (particularly the non-human world) around them [2–4].

**Funding:** Funding for this project was provided by a Gund Institute for Environment COVID-19 Rapid Research Fund award (RKG; no number assigned). Funder Name: Gund Institute for Environment (https://www.uvm.edu/gund). The funders had no role in study design, data collection and analysis, decision to publish, or preparation of the manuscript.

**Competing interests:** The authors have declared that no competing interests exist.

Shifts in human-nature interactions may potentially change how nature benefits us and how we value nature [5, 6]. Disasters may bring about such shifts, as evidenced by the increased importance of the psychological, physical, and social benefits provided by parks after Hurricane Katrina [7]. Yet for the most part, we lack a detailed understanding of how large-scale social-ecological upheaval impacts the values and benefits associated with human-nature relationships. With populations worldwide facing increased vulnerability to disasters [8, 9], we need to look more closely at how nature's values and benefits respond to and possibly help mitigate crises like the COVID-19 pandemic (hereafter, "COVID-19") [10, 11]. While a rapidly emerging body of research reveals increased activity in nature in response to the pandemic, [4, 12, 13], less attention appears directed towards the underlying values and benefits associated with this increased activity. Such knowledge could broaden our understanding of the complexity of, and potential for change within, human-nature interactions and values, as well as help improve disaster planning and management [14].

Human-nature relationships benefit people in many ways. Such benefits can be material (e.g., food, flood protection) or nonmaterial (e.g., mental health, spiritual fulfillment) [15], and can occur through human-nature interactions as diverse as subsistence practices, care and stewardship, and recreation [16–18]. These different types of benefits, the ecosystem services that provide them, and the relationships that grant humans access to them often co-occur on the same landscape and interact with each other [19]. Evidence shows that nonmaterial benefits, commonly characterized as cultural ecosystem services (CES) [9] or nature's nonmaterial contributions to people [20], contribute significantly to individual and collective quality of life [9, 21]. These contributions impact many dimensions of public health and well-being, from obvious (physical and mental health) to more subtle (e.g., sense of security, personal and community identity) [22].

The value people place on interactions with nature and relationships that confer the benefits mentioned above may lie at the heart of human behavior and concern for the non-human world [17]. Yet despite their importance, many nonmaterial benefits and values of nature remain understudied and often neglected in management decisions, taking a backseat to more easily quantifiable and generalizable material contributions [23, 24]. Overlooking these important values may limit the effectiveness of policies and programs and result in unforeseen impacts on well-being. One particular area in which these values may be relevant is the intersection of natural resource management and public health planning for disasters.

Thus far, studies that explore how disasters impact nonmaterial values largely focus on "natural disasters"–those that cause significant ecosystem changes (e.g., volcanoes, tornadoes, and fires) [25]. This leaves a need to further explore the relationship between crises that are not clear cut "natural disasters" and nonmaterial values. Why, and how, might relationships with nature change during such crises? One reason is that nonmaterial benefits and values (and some material benefits and values) do not exist "out there" but rather arise from human co-production, or the perception of importance by people under different social conditions [16]. Thus, as people's interactions with nature, or the contexts in which they interact, change, so may the ecosystem services they view as important and the values they assign those services [26].

Nonmaterial benefits and values, because they are both socially and experientially situated, are highly context-dependent [27]. Not only do nonmaterial values differ between ecosystem types; they also differ between people valuing the same ecosystem [28]. The use and value of nature can vary depending on one's demographics (e.g., age, gender, socio-economic status), cultural heritage, and experiences in nature [29–31]. Given the heterogeneity of nonmaterial benefits, distribution and access are not always evenly distributed and can be inequitably constrained by social processes [32]. Access to natural resources for subsistence and recreation,

for instance, can create conflict along economic and urban-rural gradients [33]. Since disasters are known to aggravate existing inequities [34], scholars and practitioners should much more strongly consider the connections between disasters and environmental justice [35], including how response efforts impact the nonmaterial benefits and values that different populations receive [7].

This study seeks to better understand the importance of, and characterize changes within, human-nature relationships in response to disasters by examining the impact of COVID-19 on residents of the state of Vermont in the United States of America (U.S.A.). Specifically, we report how COVID-19 changed the type of nature activities Vermonters engaged in, the frequency of that engagement, and the values associated with those interactions. Further, we explore how use, access, and value changes may differ according to socio-spatial and socio-economic status, potentially resulting in injustice. In this exploratory study, we aim to advance theory and practice in two ways: 1) we address how nature's benefits and values may change in response to disasters; and 2) we offer insights to help decision-makers create evidence-based plans and policies to sustain important human-nature relationships, and the values they confer, in current and future crisis situations.

## Materials & methods

### Data collection

We implemented our survey online via the LimeSurvey platform (S1 Dataset). We obtained Institutional Review Board approval from the University of Vermont (IRB protocol 00000913) to include survey responses from informed Vermont residents over the age of 18 to participate in the survey with a waiver of documentation of consent, submitted from their home computers or personal devices. The survey ran online through the final two weeks of Vermont's "Stay Safe, Stay Home" Executive Order (3 May– 19 May 2020) [36]. Recruitment was rolling, and simultaneous with the survey period. We recruited participants via convenience sampling; the use of non-probability sampling methods, which allow quick data collection to capture in-the-moment perspectives, is common in COVID-19 research [37]. Our recruitment strategies were: (1) two rounds of paid advertisements in Front Porch Forum, a community-level listserv that reaches about 70% of Vermont households [38]; and (2) listservs and social media platforms of 23 community partners (environmental and social-service organizations, both non-governmental and governmental). The language we used for recruitment follows. **Title:** "Vermonters, Nature, and COVID-19: A UVM Survey (paid ad)"; **Body Text:** "Does experiencing nature figure into your life right now? If so, how? Complete a brief research survey for a chance to win $50, and to help inform decisions that account for the role nature may have during events like COVID-19. Must be 18+ to participate. Learn more and take the survey".

The survey included a combination of optional open and closed-ended questions. It took, on average, 16 minutes to complete. During Vermont's initial period of restricted activity in response to COVID-19, we asked participants about their level of engagement in outdoor activities, relative to the same time last year, and their values associated with nature. We asked people about 15 specific activities, selected based on categories put forth in the *Vermont State-wide Comprehensive Outdoor Recreation Plan* and modified to reflect seasonal (e.g. we did not survey winter sports engagement) and contextual (e.g. language choices for easy comprehension by respondents) factors [39]. For each activity, we asked whether respondents engage with that activity at all, and if so, if they engaged more, less, or the same during COVID-19 (measured via a seven-point Likert scale with "no change" as the central point). We also asked people about 13 specific benefits from or values of nature. We chose these based on commonly studied CES categories [31, 40], combined with our understanding of the values and benefits

most relevant to Vermont's context. We included two material benefits (food and exercise) in this list in recognition of the interplay between nature's material and nonmaterial contributions to human well-being. Participants used a seven-point Likert scale to rate how important each of the 13 benefit/value variables (hereafter value variables) were to them during COVID-19. Then, they selected and ranked up to three benefits that were particularly strong during the pandemic. We also collected demographic data (age, gender, race, ethnicity, household income, employment status during the pandemic, and ZIP code). **Table 1** compares the aggregated socio-demographics of our sample to the Vermont population. Our sample is broadly similar to the Vermont population, we discuss over- and under- representation of certain demographics in the Limitations section.

## Data processing

To classify participants by geographic location, we used a geocoding tool in ArcGIS Pro 2.5.0 [44]. Based on the centroid of the reported ZIP codes, we created a set of X, Y coordinates for each participant. Then, we spatially joined the data with the U.S.A. Census Bureau's urban-rural classification shapefile (based on the 2010 Census) to categorize each participant as living

**Table 1. Demographic characteristics of survey respondents, tabulated for comparison to the 2014–2018 U.S.A. Census Bureau's American Community Survey [41–43].**

| Characteristic | Sample | U.S.A. Census |
|---|---|---|
| **Mean age** | 54.7 years old | 50.5 years old |
| **Gender** | | |
| Female | 63.2% (2,013) | 50.7% |
| Male | 35.8% (1,139) | 49.3% |
| Non-binary | 1.0% (32) | |
| **Race** | | |
| American Indian or Alaskan Native | 0.4% (13) | 0.3% |
| Asian | 0.5% (15) | 1.7% |
| Black or African American | 0.1% (2) | 1.3% |
| Middle Eastern or North African | 0.1% (3) | |
| Two or more races | 3.3% (105) | 1.9% |
| White | 91.6% (2,936) | 94.3% |
| Other | 4.1% (130) | |
| **Ethnicity** | | |
| Hispanic or Latino | 0.4% (13) | 1.9% |
| Not Hispanic or Latino | 99.6% (3,191) | 98.1% |
| **2019 Household Income** | | |
| Less than $10,000 | 1.2% (35) | 4.9% |
| $10,000-$24,999 | 7.2% (217) | 14.7% |
| $25,000-$49,000 | 18.4% (556) | 22.1% |
| $50,000-$74,999 | 22.3% (673) | 18.8% |
| $75,000-$99,999 | 18.9% (572) | 14.0% |
| $100,000-$149,999 | 20.3% (613) | 15.3% |
| $150,000-$199,999 | 6.7% (203) | 5.1% |
| Greater than $200,000 | 5.1% (154) | 5.0% |
| **Zip Code within Census urban-rural classification** | | |
| Urbanized Area | 26.0% (823) | 17.4% |
| Urban Cluster | 25.9% (820) | 28.2% |
| Rural | 48.2% (1,528) | 54.4% |

in an Urbanized Area (50,000 or more people), an Urban Cluster (at least 2,500 but fewer than 50,000 people), or a Rural Area (all other areas). For all other statistical analyses, we used SPSS (Statistics Package for the Social Sciences, Version 26, IBM 2019). All data processing and subsequent analysis took place at the University of Vermont.

## Data analysis

**Analysis of activities.** To understand whether the pandemic's impact on engagement with nature differed according to socio-demographic characteristics, we performed multinomial logistic regressions to identify socio-demographic patterns for the six most common activities among respondents: gardening (91% of participants); hiking (86%); relaxing alone (91%); relaxing with others (88%); walking (95%); and wildlife watching (89%) (**Fig 1**). Below the 85% threshold, there was a large decline in the proportion of our sample who engaged in each activity (**Fig 1**). In selecting activities to model, we used the most common activities because we were interested in the impact of COVID-19 on access and the possible equity dimensions of this impact, and we considered the more-widely-engaged-in activities the likeliest to show evidence of socio-demographic differences.

We performed a separate multinomial logistic regression for each of the six most common activities. Our dependent variables had three levels to indicate whether during the early phases of COVID-19 participants who engaged in a given activity did so less, the same, or more. We excluded participants who did not engage in a given activity from the model for that activity. Our models further excluded cases with missing responses for any of the independent variables. Because we wished to assess differences between three un-ordered outcomes (respondents reporting both "less" and "more" activity engagement relative to "no change"), the final data were best analyzed with a multinomial logistic regression [45]. To allow for clear analysis, we converted the data into nominal form: we condensed the three Likert scale points reflecting degrees of "less" and "more" activity engagement into one "less" and one "more" category and used responses of "no change" as the third category. For all regressions, we used $\alpha = 0.05$ to determine whether a variable was significant in the model and if there were significant

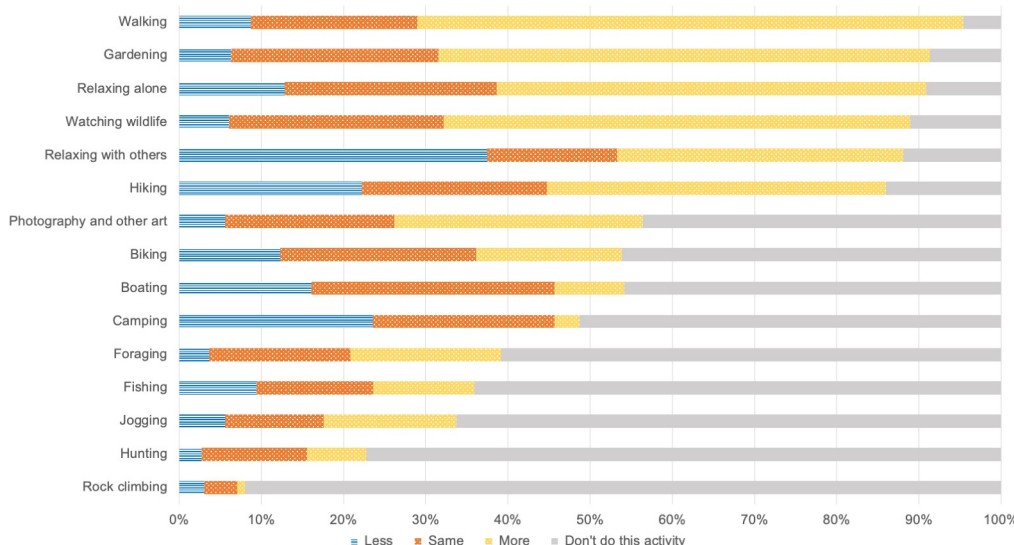

**Fig 1. Engagement in each activity during COVID-19, segmented by reported change in engagement relative to the same time last year.**

**Table 2. Independent variables and levels used in the multinomial logistic regressions.**

| Variable | Type | Levels |
|---|---|---|
| Geography | nominal | (1) urban: respondents in Urbanized Areas; (2) rural: respondents in Urban Clusters and rural areas |
| Gender | nominal | (1) female; (2) male; excludes the <1% of respondents who identified as non-binary |
| Race/ethnicity | nominal | (1) White; (2) all races/ethnicities and combinations thereof other than White |
| Income | ordinal | (1) below, (2) within, or (3) above a range ($50,000–74,999) that encompasses Vermont's median income ($60,076) as of the U.S.A. Census Bureau's 2014–2018 American Community Survey |
| Employment | nominal | (1) lost job due to COVID-19; (2) retained job in any form, including furlough or reduced hours |

differences between variable levels. We assessed model fit based on pseudo R-square statistics, Pearson and Deviance chi square statistics, and classification accuracy [45, 46].

We used five socio-demographic characteristics as independent variables in each model (**Table 2**). Although our survey also gathered age data, we excluded this variable because it introduced an abundance of cells with frequencies equal to zero into our models, substantially weakening goodness of fit. To test for collinearity between independent variables, we used a multiple linear regression with numeric versions of each dependent variable as a proxy to calculate variance inflation factors (VIF). All variables' VIF values were 1.059 or lower–well below the threshold that indicates problematic collinearity in a multivariate analysis [47].

**Analysis of values.** To investigate the possible presence of common factors that underlie our 13 value variables, we performed exploratory factor analysis. Specifically, we applied Principal Component Analysis to the 13 value items that participants rated using a 7-point Likert-scale (from strongly disagree to strongly agree). We used an oblique rotation (Oblimin) due to a moderate correlation (0.452) between the factors [48]. We allowed the data to inform the number of factors, and the result of two factors was unambiguous. The two factors cumulatively explain 58.0% of the variance in the data. To determine which value variables loaded on each factor, we used the following criteria: the loading on the factor is above 0.65, and the difference between the loading on the two factors is greater than 0.15. As an example, if loading for one factor was 0.70 but the loading for the other factor was within 0.15 of 0.70 (e.g., 0.57), we considered the variable to load similarly on both factors, so did not assign it to either factor. We characterized the two factors based on similarities among the values they encompassed. Factor 1, which included beauty, exercise, leisure, and familiarity, was characterized as "Nurture and Recreation". Factor 2, which included creativity, food, life lessons, and tradition, was characterized as "Inspiration and Nourishment". We use these characterizations throughout to identify the two factors, and investigate them in further detail in the results and discussion sections.

To identify potential socio-demographic patterns among participants who identified values from one factor as more important during the pandemic, we performed an additional multinomial logistic regression. Regressing survey participants' prioritization of factored values against their socio-demographic characteristics allowed us to investigate whether certain kinds of people experienced the pandemic's impact on values from nature differently. Similar to our rationale for applying multinomial logistic regression to our activity data (as described above in "Activities Analysis"), we hoped to address both management and equity elements of the pandemic's consequences for Vermonters' experiences of values around nature with this analysis. To accomplish this, we compared participants based on a post-hoc nominal variable with

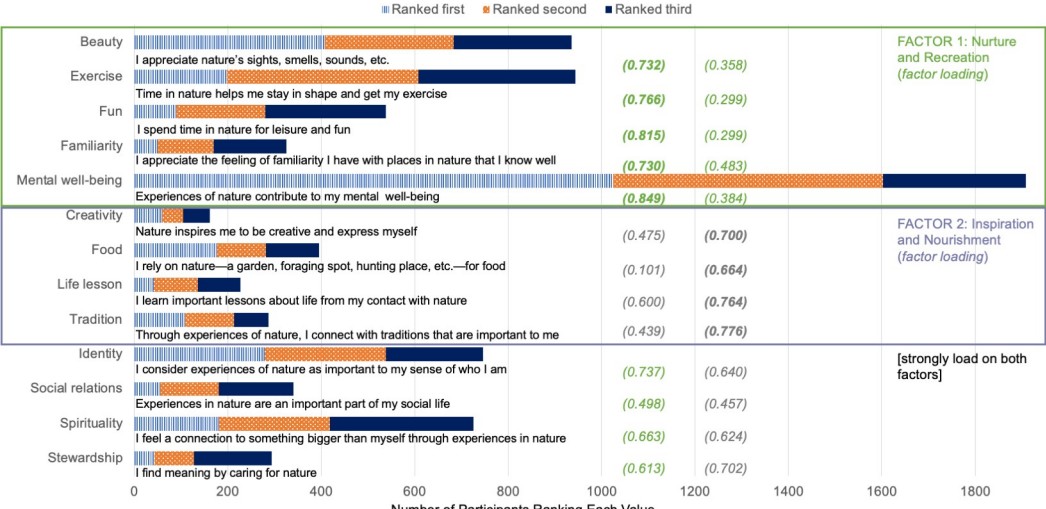

**Fig 2. Number of participants who ranked each value as most important during the COVID-19 restrictions, segmented by how participants ranked each value (as most, second-most, or third-most strongly valued).** Values are organized according to the results of the factor analysis, with the loading summarized in parentheses for each value onto one of two factors (the "Nurture and Recreation values" factor and the "Inspiration and Nourishment values" factor described in Methods).

three levels: (1) the participant ranked Nurture and Recreation values, or a combination of both Nurture and Recreation values and unassigned values (n = 1776 across the sample); (2) the participant ranked combinations of items that were not exclusive to either factor (n = 801 across the sample, **reference level**); and (3) the participant ranked Inspiration and Nourishment values, or a combination of both Inspiration and Nourishment values and unassigned (n = 136 across the sample). Because unassigned values loaded onto both factors (Fig 2), we considered them neutral with respects to a given participant's prioritization of Nurture and Recreation or Inspiration and Nourishment values. We excluded from our analysis participants who ranked no values as more important during the COVID-19 restrictions (n = 491 across the sample). Our models further excluded cases with missing responses for any of the independent variables. Table 3 summarizes the groupings of respondents by the combinations of factored and unassigned values that they prioritized as more important during the

**Table 3. Allocation of respondents to factor dependent variables for values multinomial logistic regression.**

| Combinations of values ranked as more important during the pandemic | Nurture & Recreation (Factor 1) Dependent Variable | Inspiration & Nourishment (Factor 2) Dependent Variable | Neither Factor Dependent Variable |
|---|---|---|---|
| Only Nurture & Recreation values | 584 | 0 | 0 |
| Nurture & Recreation and Unassigned values | 1192 | 0 | 0 |
| Only Inspiration & Nourishment values | 0 | 23 | 0 |
| Inspiration & Nourishment and Unassigned values | 0 | 113 | 0 |
| Nurture & Recreation and Inspiration & Nourishment values | 0 | 0 | 425 |
| Nurture & Recreation, Inspiration & Nourishment values and Unassigned values ranked | 0 | 0 | 342 |
| Only Unassigned values ranked | 0 | 0 | 34 |
| No values ranked | 0 | 0 | 491 |
| **Totals** | 1776 | 136 | 1292 |

pandemic. For independent variables, we used the same five socio-demographic variables as in our models for activity engagement (**Table 2**). For both regressions, we used α = 0.05 to determine whether a variable was significant in the model and if there were significant differences between variable levels.

## Results

We received 4,826 responses. Incomplete responses (n = 1,456) and responses that did not fit the study criteria of 18 years of age and Vermont resident status (n = 166) were discarded, for a final sample of 3,204 valid responses. The majority of participants (54%) first found this survey on Front Porch Forum, followed by 21% who found the survey from an organization's communications, 5% who received the survey from a friend or colleague, and 20% who found it from other sources (e.g., social media, news stories). The majority of participants identified as White (92%). Participants lived in rural areas (48%), urban clusters (26%), and urban areas (26%). The average age was 54.7. Median reported household income was $75,000 - $99,999. Women comprised a small majority of our sample (63%) (**Table 1**). Participants' pandemic job situations varied, though the plurality (48%) continued to work from home with hours unchanged.

### Activities

Participants collectively reported engagement in all 15 nature-based activities included in the survey (**Fig 1**). Nearly all participants (95%) reported walking; the smallest proportion (8%) reported rock climbing. Changes in engagement relative to the same time last year varied. Activities with the largest proportion of individuals reporting a similar level of engagement to the same time last year were biking (44%), boating (55%), fishing (40%), hunting (56%), and rock climbing (51%). The largest proportion of individuals reported increased engagement for foraging (47%), gardening (57%), hiking (48%), jogging (48%), photography and other art (54%), relaxing by myself (58%), walking (70%), and wildlife watching (64%), and decreased engagement for camping (48%) and relaxing with others (43%).

Multinomial logistic regression results (**Table 4**) provide insight into the socio-demographic factors associated with increased engagement in selected activities: gardening; hiking; relaxing socially; relaxing alone; walking; and wildlife watching. All models were statistically significant (p < 0.05), with low Pseudo R-square measures, and with Pearson and Deviance chi square values in agreement for good model fit (p > 0.05) for all activities except for Gardening (Pearson > 0.05, Deviance p < 0.05). This discrepancy may derive from the proportion of cells with zero frequencies in the Gardening model, 23%; all other models had <20% cells with zero frequencies, per literature recommendations [45]. We therefore interpret the results of our Gardening model with caution. Classification accuracy revealed that our models were reliable only when predicting odds of membership in the largest levels of each dependent variable (increased activity), except in the case of Relaxing Socially, which exhibited more even counts across dependent variable levels. We calculated the proportional-by-chance accuracy rate of classification for each model, with a 25% improvement over chance as our threshold [45]. All models were accurate by this measure for predicting odds of a participant having increased activity engagement.

### Values

The two factors that the principal component analysis identified encompassed five and four values variables, respectively. **Fig 2** presents factor loadings, organized by factor. We named

**Table 4. Demographic variables that showed significantly higher odds for increased activity engagement compared to no change and were significant in the model (* = p<0.05; ** = p<0.01; *** = p<0.001).**

| Independent Variables (Odds ratios for levels significantly more likely to report increased activity engagement; Relaxing Socially includes odds for decreased engagement as well. | Geography | Gender | Race/ Ethnicity | Income | Employment |
|---|---|---|---|---|---|
| **Gardening** (n = 2,521; $X^2$ = 72.54; df = 12; p<0.001; Pseudo R-square = 0.028) | Rural*** (1.28)* | Female*** (1.67)*** | | | Lost job* (1.72)** |
| **Hiking** (n = 2,387; $X^2$ = 80.31; df = 12; p<0.001; Pseudo R-square = 0.033) | | Female*** (1.70)*** | | | |
| **Relaxing Socially Increased** (n = 2,448; $X^2$ = 123.45; df = 12; p<0.001; Pseudo R-square = 0.049) | | Female*** (2.82)*** | | Above*** median (1.44)* | Lost job* (1.75)* |
| **Relaxing Socially Decreased** (as above) | Urban** (1.08)* | Female*** (1.21)*** | | | |
| **Relaxing Alone** (n = 2,518; $X^2$ = 93.21; df = 12; p<0.001; Pseudo R-square = 0.036) | | Female*** (1.94)*** | | | |
| **Walking** (n = 2,650; $X^2$ = 164.17; df = 12; p<0.001; Pseudo R-square = 0.06) | Urban** (1.32)* | Female*** (2.85)*** | | | Lost job* (1.63)* |
| **Wildlife Watching** (n = 2,453; $X^2$ = 113.39; df = 12; p<0.001; Pseudo R-square = 0.045) | | Female*** (2.08)*** | | | Lost job** (1.67)** |

We inverted some odds ratios and report corresponding independent variable levels for ease of interpretation.

1. The first result can be read as: "people in rural areas had higher odds of reporting increased gardening."

2. Pseudo R-squares reported here are Cox & Snell.

3. Dependent variables and model fitting information pertain to each row.

4. For income, below median is the reference level.

5. Full summaries of each regression are in supplemental materials S2–S7 Tables.

Factor 1 "Nurture and Recreation values" and Factor 2 "Inspiration and Nourishment values" to summarize the essence of each combination of value variables.

Multinomial logistic regression revealed that several socio-demographic factors were associated with increased odds of participants' rankings of Nurture and Recreation values as more important during COVID-19 (**Table 5**). The model was statistically significant, exhibited low Pseudo R-squares, had acceptable Pearson and Deviance chi-square statistics (both with p > 0.05), and, as in our activity regressions exhibited accurate classifications only for the largest level of the dependent variable (Nurture and Recreation). Also as in our activity regressions, we calculated the proportional-by-chance accuracy rate of classification for odds of

**Table 5. Demographic variables that were significant in the model (* = p<0.05; ** = p<0.01; *** = 0<0.001), and showed significantly higher odds for ranking nurture & recreation values as more important during COVID-19 compared to unassigned values.**

| Independent Variables (odds ratios for levels significantly more likely to rank a Nurture & Recreation factor as more important) | Geography | Gender | Race/ Ethnicity | Income | Employment |
|---|---|---|---|---|---|
| **Factor 1 Values** (n = 2,367; $X^2$ = 95.26; df = 12; p < 0.001; Pseudo R-square = 0.039) | Urban*** (1.79)*** | Female*** (1.69)*** | | above median** (1.46)**; (1.28)* | |

We inverted some odds ratios and reported corresponding independent variable levels for ease of interpretation.

1. Pseudo R-squares reported here are Cox & Snell.

2. Dependent variables and model fitting information pertain to each row.

3. For income, below median is the reference level; odds ratios for above median and median are reported in that order.

4. Full summaries of each regression are in supplemental materials S1 Table.

membership in the Nurture and Recreation level, and exceeded a 25% improvement over chance [45].

## Discussion

During the first few months of the COVID-19 pandemic in the U.S.A., many people's relationships with nature changed [2, 4]. As government restrictions and individual responses limited mobility and social relations, people interacted with nature differently—sometimes more, sometimes less, and sometimes in new ways. Simultaneously, the values people associate with nature morphed. These changes have implications for research on human-nature relationships, environmental values, and the role of socio-demographic factors in human-nature relationships and values.

Our snapshot of one place, the U.S.A. state of Vermont, demonstrates the kinds of changes that likely happened in many places. Within these shifts, we observed multiple patterns based on people's backgrounds. These observations matter because human-nature relationships support individual and community well-being [9], which is especially vulnerable during disasters like COVID-19 [4, 49]. Research demonstrates that exposure to natural environments supports recovery from stress [50–52] and suggests that these relationships may provide a source of resilience for both individuals and communities experiencing disasters [10]. However, disasters impact different people in different ways [35], and understanding this variability could help future disaster and conservation planning efforts to optimally help people—particularly across diverse communities.

Building on this literature, several studies have begun to document the effects of COVID-19 on nature experiences and their value to people. Preliminary evidence shows that both visits to and the value of natural areas have increased during the pandemic [2, 53], and that compared to time spent indoors during COVID-19, time spent outdoors was associated with greater psychological well-being [3] and decreased feelings of loneliness during lockdown conditions [54]. However, most research to date has not thoroughly characterized changes in specific values from or in relation to nature in response to the pandemic, or placed these values in dialogue with shifting levels of engagement in nature activity. Current research at a broad geographic scale is limited to a focus on recreational values [4], while research that examines more diverse values does so in the context of geographically specific place relationships [2]. Our study adds to this work by examining how people's nature-based activities and the values associated with them have changed at an intermediate geographic scale and with regard to diverse values, as well as by relating changes in engagement in outdoor activities to socio-demographic differences. We hope this contribution offers a launching point for discussion of the interplay between diverse values and wide-ranging forms of engagement with nature as sites to examine the potential for change in human-nature relationships.

### COVID and nature-based activities

Though some people reported changes in their engagement in nature-based activities during COVID-19 compared to the same time last year, others reported increased or decreased participation in specific activities. The potential for engagement in nature activities to change across a large population hints at one pathway by which individual and collective human-nature relationships might evolve in response to crisis. Wide-ranging cultural practices contribute to the norms and expectations that shape communities' experiences of nature [16]. In the context of contemporary, Western natural resources research and policy, instrumentally valued recreational relationships are often prioritized among these activities [18]. Yet a broad literature recognizes diverse activities, including caring and reciprocal practices, as equally central across

many cultural contexts [55, 56]. Here, we discuss changes in participation in a wide range of recreational and non-recreational activities.

For most activities we surveyed, there were more reports of increases in activity engagement than reports of unchanged levels or decreases in engagement. A plurality of respondents reported decreases in camping and relaxing with others, and though most individuals engaging in boating and rock climbing reported no change, those reporting a decrease were more numerous than those reporting an increase. Many factors might explain these changes. Closures during the early stages of the pandemic limited access to recreational areas and transportation options that enable some of these activities. People also may have responded to calls within the outdoor-recreation community to reduce potentially risky activity. A Vermont-based nonprofit dedicated to rock climbing, for instance, discouraged participation during the state's lockdown period to avoid possible exposure to the virus and injuries that could burden medical resources. Further, some people probably made personal decisions to avoid activities that involve other people or crowds. In comparison, activities for which respondents primarily reported maintaining the same level of engagement rather than a change—biking, boating, fishing, and hunting—can usually be done alone. The possible exception here is rock climbing, though it also can be done alone or with just one partner, and in rather secluded locations.

Where increased activity engagement was concerned, a plurality of respondents reported more frequently foraging, gardening, hiking, jogging, engaging in photography or art, relaxing by themselves, walking, and watching wildlife. Relatedly, among activities where a plurality of respondents reported no change, more individuals reported increases rather than decreases for biking, fishing, and hunting. Again, many potential factors may explain these increases. Activities with a high rate of respondents reporting increases compared to decreases, like walking and watching wildlife, have relatively low barriers to participation and can be done almost anywhere [57]. Smaller increases in other activities are perhaps surprising considering the potential barriers to participation, such as time, cost, opportunity, knowledge, and ability [39]. Participation in fishing and hunting, for instance, not only requires specialized equipment and skills but also relies on cultural meaning and social support [58]. Other activities, like biking, hiking, and jogging, can be low-cost, but carry certain stereotypes, benefit from mentoring, or raise concerns related to safety in ways that may exclude certain populations [59, 60].

We found several significant socio-demographic trends associated with increased activity engagement. First, the odds of reporting increased gardening, hiking, relaxing socially, relaxing alone, walking, and wildlife watching were higher for female respondents. This finding is striking: female respondents were the only demographic who reported increased activity across all six of the most engaged-in activities we surveyed. The higher odds of increased engagement in a plethora of nature activities by women contribute to the understanding of gender differences during the pandemic. Although research on the gender equity implications of COVID-19 is presently limited [61], the pandemic has likely increased professional and household burdens on women much more than men [62]. How this finding interacts with our gender-related findings is a rich area for future study; for instance, women may have had a greater need for stress relief during the pandemic, and are potentially more likely than men to turn to nature for stress relief.

Second, the odds of reporting increased gardening, relaxing socially, walking, and wildlife watching were higher for respondents who had lost their jobs during the pandemic than those who retained them. In some respects, increased outdoor activity engagement by unemployed respondents is not surprising; unemployment results in less structured time, and outdoor activities provide a well-documented source of stress relief [63, 64]. However, this finding does offer a potential rebuttal to arguments that nonmaterial benefits from engagement with nature, such as stress reduction and social connection, are "luxury goods" [30]. Even during crises that

threaten access to material goods, respondents who lost their jobs due to the pandemic are prioritizing a range of outdoor activities is suggestive of the wide range of benefits that may arise from diverse activities in nature [16]. While increased odds of gardening could be attributable to offsetting one material stressor associated with job loss (food security), increased odds of walking, relaxing outside with others, and wildlife watching suggest that respondents who lost their jobs are seeking more diverse benefits from engagement with nature, even at a time of material crisis. It is also informative to consider this result alongside the related finding that respondents earning above-median incomes also increased engagement in relaxing outside with others during the pandemic. Given the likelihood that above-median earners retained their jobs, these paired findings hint at the importance of outdoor activity as a source of social connection that spans the socio-economic spectrum.

Third, the odds of reporting increased gardening were higher for rural respondents, while the odds of reporting increased walking were higher for urban respondents. That geography would impact activity outdoor activity engagement is not surprising. As food insecurity has increased in Vermont during the pandemic [65], residents may be turning to self-production to supplement their resources [personal correspondence, Dr. Meredith Niles]. As gardening is limited by available space, rural respondents in our sample may have had greater access to places to garden than urban respondents. Likewise, urban respondents' higher odds of reporting increased walking may reflect the relatively limited outdoor activities available in a city context. National surveys reveal decreased outdoor recreation by urban residents in terms of frequency, duration, and distance traveled [4], and suggest that travel for outdoor recreation may be mediated by risk tolerance [66]. In this light, increased walking adds to our understanding of urban respondents' response to the pandemic by revealing that outdoor activity engagement may not be uniformly decreasing; rather, it may be shifting to more accessible activities with reduced associated travel risk, consistent with shifts in geographic scale of other outdoor recreational activities [67].

Finally, it is noteworthy that our model for relaxing socially outdoors accurately predicted odds for both increased and decreased engagement—the only model in our analysis to do so. This signals less uniform skew in the direction of nature engagement due to the pandemic for this activity. It makes sense that we might observe a more nuanced trend for relaxing socially outdoors. Outdoor gathering is widely recognized as safer than indoor gathering during the pandemic, which could explain increases in this behavior. At the same time, risk averse individuals may be prioritizing safety over any form social gathering, even outdoors. The relatively small odds ratios of decreased activity (1.08 for urban respondents; 1.21 for female respondents) for relaxing socially outdoors suggest a less pronounced effect than many of the trends towards increased activity that we report.

## COVID and values from nature

In addition to changes in activities, people reported that they not only highly valued the benefits of human-nature relationships during COVID-19, but that some benefits and values were more important than others during the first few months of the pandemic. While wide-ranging research establishes activities in nature as a key element of human-nature relationships, the values emerging from those connections play an important role in understanding behaviors and broader orientations that characterize such relationships at both individual and societal scales [16, 68]. Indeed, the rapidly growing literature on relational values recognizes that human-nature relationships can be valued in their own right, in ways distinct from the instrumental and intrinsic value framings common in Western research [69, 70]. Thus, evidence of value change in response to the pandemic is a second potential pathway to increase our understanding of human-nature relationships, including how they may change.

The values people said were most important factored into two distinct groups: Nurture and Recreation values (factor 1) and Inspiration and Nourishment values (factor 2). Each factor included both material and nonmaterial elements. Empirical research demonstrates that people discuss ecosystem services (ES) and values in intertwined ways [71]—that people tend to "bundle" multiple services, and that these bundles can be associated with both material and nonmaterial values [19]. The limited number of studies that explore how nonmaterial ES relate to each other find that they often interconnect and bundle together [72, 73]. Rojas et al., for example, analyzed ES post-disaster (an earthquake) using factor analysis and found four distinct factors of nonmaterial ES: (1) mental recreation and appreciation of flora and fauna; (2) social relations, traditional knowledge, and aesthetic value; (3) sports and eco-tourism; and (4) scientific activity and environmental education [74]. Additionally, the limited research on bundling and demographics suggests that socio-demographic factors may be related to bundling [75, 76]. Martín-López et al., for instance, found that valued CES varied between urban and rural people in Spain: urban residents highly valued aesthetics, tourism, environmental education, and existence value, whereas rural people highly valued recreational hunting and local ecological knowledge [73]. Understanding how socio-demographic factors like urban/rural divides relate to the bundling of benefits and values could be crucial; as populations around the world (including in the U.S.A) becoming more urban, the economic, social, and political gaps between urban and rural populations continue to grow. This puts pressure on decision-makers to meet the needs of increasingly diverse citizens [77].

Our study adds to the growing evidence that non-material ES interact and cluster together, along with material ES (e.g., food and physical exercise). It also supports the idea that these bundles are shaped by their ecological and social contexts. Unlike Rojas et al. [74], in our data, mental well-being and aesthetic value are bundled, and cultural heritage falls into a separate factor. We found that urban respondents were more likely to rank Nurture and Recreation values (mental well-being, aesthetics, fun, exercise, and familiarity) as more important during the early months of COVID-19. This urban effect in part matches previous research [73] but modifies the elements of observed value bundles with the additional prioritization of mental well-being by urban respondents. These findings suggest that value heterogeneity can exist in a relatively small geographic area (Vermont) and within a relatively homogenous population (Vermont's mostly White, older-skewing, population). In a largely rural state like Vermont, even relatively small urban areas, like Burlington (population 42,899), are seen as (and shown in our data to be) culturally distinct [78].

Gender and income also influenced the likelihood that participants prioritized Nurture and Recreation. Women and respondents with median- and above-median incomes were more likely to rank Nurture and Recreation values as more important than men and respondents with below-median incomes. The greater likelihood that women would rank Nurture and Recreation values as more important might be explained by the effect of gender roles and socialization on mental well-being—women are more likely to report (and presumably by extension, highly rank survey items about) emotional concerns (e.g., mental well-being) [79]. Similarly, the Nurture and Recreation value factor's focus on leisure and exercise may explain its alignment with higher-income participants, since these individuals are more likely to engage in physical activity as leisure rather than as work (paid or unpaid) [80].

Our findings on population-level differences in value change during COVID highlight equity as an important consideration for decision makers seeking to anticipate how policies may result in inequitable impacts. Though equity dimensions of access to outdoor activities have entered academic [60] and mainstream [81] consciousness, less attention has been paid to the benefits and values associated with those activities. Understanding how benefits and values vary among groups, especially during crises, could provide important insight into nature's

diverse values, as well as inform environmental justice efforts. Ensuring that all communities have access to "nature-based health amenities" can improve societal and health equity [82]. Additionally, communication efforts that focus on maintaining well-being may not currently acknowledge these value differences, because they heavily emphasize messages such as mental well-being benefits [83], which our findings suggest resonate most with median- and above-median income, urban populations. By framing nature interactions too narrowly, public health campaigns may unknowingly limit their reach and effectiveness [84].

## Limitations

Noteworthy in our analysis is the prevalence of rare events among our dependent variables [85]. In our values analysis, the number of participants who ranked Nurture and Recreation values (n = 1,776) dwarfs those who ranked Inspiration and Nourishment values (n = 136) as more important during COVID-19. The most literal interpretation of this disparity is that more people value Nurture and Recreation values than Inspiration and Nourishment values in the context of COVID-19. However, we also consider the possibility that public discourse (e.g. media and popular framing [86]) of nature's benefits during times of crisis may emphasize values like mental health or nature's beauty over values like creative inspiration or connection to heritage. Such framings would impact individual perceptions of nature's value. Alternately, the Inspiration and Nourishment values we studied may be harder for individuals to conceptualize, especially in the limited format of a quantitative survey [87]. They may also be subject to a temporal effect such that values like connections to tradition, creative inspiration, or learning life lessons might not manifest as strongly early in a respondent's pandemic experience. Reliance on nature for food, of course, could manifest immediately for some respondents.

Likewise, five of our activity regressions exhibited at least one level of the dependent variable as sufficiently small enough to be considered a rare event [85]. In these cases, the largest level of the dependent variable was for increased activity engagement. A convenient interpretation of this result would be that the pandemic has increased Vermonters' reliance on outdoor activity engagement to mediate stress; as a predominantly rural state, this interpretation of our data would not be overtly at odds with literature reporting decreased outdoor activity engagement among urban residents [4]. However, a troubling and plausible possibility is that our sampling approach may have been biased in favor of respondents who are motivated and able to respond to the pandemic through nature engagement. Our reliance on environmental organization listservs to distribute our survey, and advertising language soliciting respondents for whom "experiencing nature figure[d] into [their] life" at the time of sampling could contribute to such a bias.

Regardless of cause, the prevalence of rare events across our model dependent variables presents both analytical and research equity challenges. Though widely used for analyzing categorical data in social science and policy contexts, logistic regression is poorly suited to modeling variables in which one level is much more rare than the other(s) [85, 88]. Although bias-reduction methods to address this shortcoming exist, they are most effective for binary, rather than multi-level, response variables [89, 90]. The most promising statistical package we found to implement one common correction used in rare events logistic regression (the "Firth" correction) regression remains a work in progress where multinomial models are concerned [91]. Our trial application of this package did not result in more accurate rare events classification than the un-corrected regressions, which is not surprising based on other studies [90]. Approaches to justify interpretation of only the larger dependent variable level in a rare events multinomial logistic regression exist; we applied these as described in our methods [45]. However, doing this prevented us from commenting on socio-demographic trends that might be at

play among respondents who may be considered marginalized—that is, respondents whose outdoor activity engagement was decreased rather than increased by the pandemic, and respondents who prioritized the less common Inspiration and Nourishment values. Thus, while our findings accurately report the dominant trends in outdoor activity engagement and associated values during the pandemic in Vermont, we run the risk of further silencing communities whose pandemic experiences do not mirror the majority's without carefully acknowledging this shortcoming in our policy recommendations (see below).

Three additional limitations should be considered when interpreting these results. First, although they are statistically significant, our multinomial logistic regression models have low pseudo R-squares. This statistic is one way to estimate model fit [45]. With this in mind, factors not accounted for in our models are likely important contributors to respondents' outdoor activities and prioritization of values during the pandemic. The second concern is the possibility that some effects we observed might be influenced by circumstances other than COVID-19. Our wording of activity engagement questions asked participants to compare their levels of activity during the pandemic to those one year before. Accordingly, a non-COVID event (e.g., injury, development of a new hobby, birth of a child) could explain changes in people's activities and values. In aggregate, however, we can assume that most changes were COVID-induced.

The third, more substantial, limitation is that our study used a non-random convenience sample of internet users, which further restricts generalizability. Chief weaknesses of this method include the risk of generating a non-representative sample of the study population, and the risk of selection biases among respondents [92]. Noteworthy in this respect are the higher proportion of women than men in our sample (which is consistent with user trends for Front Porch Forum, the main platform we used for recruitment [68]) and the skew towards higher income ranges among our respondents. In an effort to reduce some of the biases common to convenience sampling, we offered participants the chance to enter a drawing for a $50 gift card, contingent upon completing both the initial survey (which serves as the basis for this paper) and a follow-up survey for planned longitudinal research. This measure may have helped encourage more diverse participation from within the population, but would not have controlled for self-selection bias originating in the advertising platforms, and the nature-specific language used in our posting (e.g. outdoors enthusiasts may be over-represented relative to the Vermont population as a whole). At the same time, the study design allowed us to rapidly and cost-effectively distribute the survey and collect a large number of responses from across the population, consistent with the acknowledged strengths and common applications of this method [93]. With these considerations in mind, we recommend caution in generalizing our results beyond rural predominantly rural areas in the Northeastern United States. Although our sample resembles the overall Vermont population in most respects, the state is small and quite homogenous compared to many other locations within the U.S.A.

Finally, many questions remain as to the nuances of changes in people's relationships to nature during the pandemic, and whether these changes will be long-lived. Our survey did not capture whether respondents took up entirely new activities as a result of the pandemic. Given reports of activity changes, such as a surge in Vermont hunting license and fishing license sales, future surveys should ask about engagement in new activities, as well as the continuation of those new activities post-restrictions. Similarly, we could not determine if the values people experienced as more important represented previously held or completely new values, nor if these changes represent a permanent shift in how people think about the importance of nature. At the time of this writing, we had recently concluded a follow-up survey with our original respondents to better understand some of these questions, including the long-term impacts of the pandemic on their relationships with nature.

## Conclusion and policy recommendations

Our study shows how the early stages of the COVID-19 pandemic in the United States impacted Vermont residents' interactions with nature and the values associated with those relationships. This knowledge adds to our growing understanding of the potential for change within human-nature relationships (if only on a very short timescale), as well as how changes related to social upheaval may vary across diverse populations. It also has practical implications for disaster- and natural resources-planning efforts. Given the links between nature's values and human well-being, and the potential role of nature's values in disaster recovery and resilience, emergency decision-makers should consider how policies enable and/or hinder important human-nature interactions. Our results lend themselves to such consideration in two areas.

First, our study provides a thorough overview of the outdoor activities seeing the greatest increases in engagement during the pandemic, and explores these shifts by socio-demographic characteristics. Natural resource agencies across the United States are facing COVID-driven budget cuts [94, 95]. Allocating limited resources to sustain crucial opportunities for outdoor activity should be a key priority—but in a way that acknowledges the diversity of activity and values priorities that can exist in even relatively small geographies, among relatively homogenous populations. As such, we urge decision-makers to seek fine-scale analysis to ensure that the most widely accessible outdoor activities receive the support needed to meet demand, for both frequent participants and marginalized populations for whom the benefits of such participation might be especially important in times of crisis. Thus, we caution that in such planning particular care should be taken to understand the barriers to engagement in these activities, and to prioritize reducing these barriers alongside merely sustaining access.

Second, our better understanding of the values associated with nature during the pandemic also could improve messaging about how to maintain well-being despite social-distancing restrictions. The range of values from nature that we report, the presence of distinct factors within that range, and the disparity in frequency of prioritization across factors, suggest that human-nature relationships can be founded on wide-ranging interests, even in shared landscapes and among stakeholder communities that seem, at first blush, homogenous [96, 97]. This should indicate to natural resource managers a need for nuanced and equally wide-ranging messaging; our findings support the existence of important and diverse values that can otherwise be overlooked. Planners should consider not just the quantity, but also the quality and spatial organization of nature access, in order to provide natural spaces that equitably support people's needs under a range of social and ecological conditions [98].

Lastly, as Dandy [99] noted, "Perhaps the most vital question is whether, as a result of the current crisis, there will be any change in the values that underpin and guide human behaviours?" Our results suggest at least some degree of value change, even if on a short timescale with uncertain permanence. Future research can explore to what extent these changes may persist, and the possible implications of those shifts on the well-being of people and nature.

## Supporting information

**S1 Dataset. Survey dataset.**
(XLSX)

**S1 Metadata.**
(PDF)

**S1 Table. Full SPSS output tables for each of the eight multinomial logistic regressions summarized in the results section and Tables 4 and 5.** These tables include: sample sizes,

model fitting information and pseudo R-squares for each regression; Pearson and Deviance chi-square statistics; classification tables; and likelihood ratio tests, coefficients (with standard error), p values, and odds ratios (with 95% confidence intervals) for each variable in each regression.
(XLSX)

**S2 Table. Full SPSS output tables for each of the eight multinomial logistic regressions summarized in the results section and Tables 4 and 5.** These tables include: sample sizes, model fitting information and pseudo R-squares for each regression; Pearson and Deviance chi-square statistics; classification tables; and likelihood ratio tests, coefficients (with standard error), p values, and odds ratios (with 95% confidence intervals) for each variable in each regression.
(XLSX)

**S3 Table. Full SPSS output tables for each of the eight multinomial logistic regressions summarized in the results section and Tables 4 and 5.** These tables include: sample sizes, model fitting information and pseudo R-squares for each regression; Pearson and Deviance chi-square statistics; classification tables; and likelihood ratio tests, coefficients (with standard error), p values, and odds ratios (with 95% confidence intervals) for each variable in each regression.
(XLSX)

**S4 Table. Full SPSS output tables for each of the eight multinomial logistic regressions summarized in the results section and Tables 4 and 5.** These tables include: sample sizes, model fitting information and pseudo R-squares for each regression; Pearson and Deviance chi-square statistics; classification tables; and likelihood ratio tests, coefficients (with standard error), p values, and odds ratios (with 95% confidence intervals) for each variable in each regression.
(XLSX)

**S5 Table. Full SPSS output tables for each of the eight multinomial logistic regressions summarized in the results section and Tables 4 and 5.** These tables include: sample sizes, model fitting information and pseudo R-squares for each regression; Pearson and Deviance chi-square statistics; classification tables; and likelihood ratio tests, coefficients (with standard error), p values, and odds ratios (with 95% confidence intervals) for each variable in each regression.
(XLSX)

**S6 Table. Full SPSS output tables for each of the eight multinomial logistic regressions summarized in the results section and Tables 4 and 5.** These tables include: sample sizes, model fitting information and pseudo R-squares for each regression; Pearson and Deviance chi-square statistics; classification tables; and likelihood ratio tests, coefficients (with standard error), p values, and odds ratios (with 95% confidence intervals) for each variable in each regression.
(XLSX)

**S7 Table. Full SPSS output tables for each of the eight multinomial logistic regressions summarized in the results section and Tables 4 and 5.** These tables include: sample sizes, model fitting information and pseudo R-squares for each regression; Pearson and Deviance chi-square statistics; classification tables; and likelihood ratio tests, coefficients (with standard error), p values, and odds ratios (with 95% confidence intervals) for each variable in each

regression.
(XLSX)

**S1 File. Survey instrument.**
(PDF)

## Acknowledgments

We thank Margaret Lee and Eliza Merrylees for their help cleaning and formatting the data from our survey, and gathering relevant literature. Thanks also to Jesse Freedman and Timothy Terway for feedback on creating the urban-rural classification, to Walter Kuentzel and Meredith Niles for feedback on research design, to Nelson Grima and Jessica Wikle for assistance with statistics, and to Alison Adams for copy-editing. Finally, we thank our community partners for their help distributing the survey.

## Author Contributions

**Conceptualization:** Joshua W. Morse, Tatiana M. Gladkikh, Diana M. Hackenburg, Rachelle K. Gould.

**Data curation:** Joshua W. Morse, Tatiana M. Gladkikh, Diana M. Hackenburg.

**Formal analysis:** Joshua W. Morse, Tatiana M. Gladkikh, Rachelle K. Gould.

**Funding acquisition:** Joshua W. Morse, Tatiana M. Gladkikh, Diana M. Hackenburg, Rachelle K. Gould.

**Methodology:** Joshua W. Morse, Diana M. Hackenburg, Rachelle K. Gould.

**Project administration:** Joshua W. Morse, Rachelle K. Gould.

**Writing – original draft:** Joshua W. Morse, Tatiana M. Gladkikh, Diana M. Hackenburg, Rachelle K. Gould.

**Writing – review & editing:** Joshua W. Morse, Tatiana M. Gladkikh, Diana M. Hackenburg, Rachelle K. Gould.

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
