## [Decision Letter · Decision Letter 0]

9 Sep 2020

PONE-D-20-23932

COVID-19 and human-nature relationships: Vermonters’ activities in nature and associated nonmaterial values during the pandemic

PLOS ONE

Dear Dr. Morse,

Thank you for submitting your manuscript to PLOS ONE. After careful consideration, we feel that it has merit but does not fully meet PLOS ONE’s publication criteria as it currently stands. Therefore, we invite you to submit a revised version of the manuscript that addresses the points raised during the review process.

We look forward to receiving your revised manuscript.

Kind regards,

Julia Martin-Ortega

Academic Editor

PLOS ONE

Additional Editor Comments:

Both reviewers are very positive about this manuscript, and I support their views on the relevance and rigour of the study. However, i do find that the manuscript in its current form doesnt suffiently rise to the promise it sets up in the introduction with respect to advancing theory and practice on the understanding of human-nature relationships. I believe, however, that can be addressed with relatively moderate revision of the manuscript, notably by further developing/depening of the discussion and the conclusions. Firstly, you need to explain what do you mean by "dynamism" of human-nature relationships and how specifically that is reflected in your study design (do you mean just change? or is it a more complex concept than that?). Secondly, the discussion mostly describes patterns of behaviour detected in the survey, but I miss a deeper/more profund discussion on what does this fundamentally means in terms of our understanding of human nature relationships. In the introduction you establish that "This knowledge could broaden our understanding of the complexity and dynamism of human-nature interactions and values, as well as help improve disaster planning and management", but most of the discussion describes the behavioural changes (which are, by the way, well discussed/described and referenced) and not so much the complexity and dynamism of such relationships and I believe this could be done more/further developed. you do have some good grounds (and you make some interesting initial discussion) when you discuss the results of the Nurture/Recreation vs Inspirational values aspects of your analysis. i would like to see that further developed with respect to what does it mean for our understanding of the complex relationships. Thirdly, and still in relation to rising to the promise made at the introduction, i find that most of the recommendations for policy making made in the conclusion section are quite generic (and somewhat well known) and could be made more specific/insightful with respect to disaster managment and in the context of the pandemic.

Besides the above, which I consider to be the more fundamental issues that need to be addressed, there are a number of other issues that i would like to see addressed in the next version of the manuscript (besides those made by the reviewers):

- The methodology section does not explain sufficiently clearly/explicitly what is the purpose of regressing the dependent variables against the factor/values identified in the factor analysis. i believe this is a key contribution of the study, and while you explain well the procedure that you follow to do this, you dont explicitly/sufficiently explain why are you doing so, what is the purpose/value of doing this. Actually, by explaining and motivating this, i believe you are going to find it easier to make more out of it in the discussion (as per my comment above). You should provide some rationale of this process in the section that you call "Analysis of values".

- It would be clearer if you defined the two values (Nurture/Recreation; Inspiration/Nourishment) when you first present the factor analysis (in the Analysis of values section) and not in the results section. it would help with clarity (to be honest, i didnt' understand what you were doing with these until quite late in the manuscript - more clarity and earlier signposting of the prupose of this part of the analysis should help with that)

- Similarly, you need to explain more specifically what the results mean/imply. In L263 you say: "Multinomial logistic regressions revealed that several 263 socio-demographic factors were associated with increased odds of participants’ rankings of either Nurture and Recreation values or Inspiration and Nourishment values as more important during COVID-19 (Table 4)". Rather than just refering to the table, it would help if you developed in the text more what are those findings and what they mean.

- In the limitations section, consider referering to self-selection bias. how was the survey presented to participants? could it be that those more active in recreational activities felt more inclined to respond to the survey? how was the framing/context of the survey? etc.

- You make some reference to Vermont being quite particular in the US content. what about the possibility of your study being more or less relevant more broadly? Do you have any info you could contextualize your results more broadly in the US/other countries (e.g. in the global north). Maybe there is not yet sufficient evidence to refer to (it's is quite early)but you could do it in relation to other related research (i.e. not COVID specific)

- In the methods section you explain that the geographical location of participants was partly recorded. this doesnt seem to have been used in the analysis (was it used to differentiate urban/rural?). Could there been room for a more fine grained spatial analysis, e.g. with relation to access to sites, etc.

In the overall, I feel positive about the potential of the research to be published if these and the reviewers' comments are addressed.

Journal Requirements:

2. Please include additional information regarding the survey or questionnaire used in the study and ensure that you have provided sufficient details that others could replicate the analyses. For instance, if you developed a questionnaire as part of this study and it is not under a copyright more restrictive than CC-BY, please include a copy as Supporting Information.

3. In your Methods section, please provide additional information about the participant recruitment method and the demographic details of your participants. Please ensure you have provided sufficient details to replicate the analyses such as: a) the recruitment date range (month and year), b) a description of any inclusion/exclusion criteria that were applied to participant recruitment, c) a table of relevant demographic details, d) a statement as to whether your sample can be considered representative of a larger population, e) a description of how participants were recruited, and f) descriptions of where participants were recruited and where the research took place.

4. Please provide additional details regarding participant consent. In the ethics statement in the Methods and online submission information, please ensure that you have specified what type you obtained (for instance, written or verbal, and if verbal, how it was documented and witnessed). If your study included minors, state whether you obtained consent from parents or guardians. If the need for consent was waived by the ethics committee, please include this information.

Reviewers' comments:

Reviewer's Responses to Questions

**Comments to the Author**

1. Is the manuscript technically sound, and do the data support the conclusions?

Reviewer #1: Yes

Reviewer #2: Partly

2. Has the statistical analysis been performed appropriately and rigorously? 

Reviewer #1: Yes

Reviewer #2: No

3. Have the authors made all data underlying the findings in their manuscript fully available?

Reviewer #1: Yes

Reviewer #2: Yes

4. Is the manuscript presented in an intelligible fashion and written in standard English?

Reviewer #1: Yes

Reviewer #2: Yes

5. Review Comments to the Author

Reviewer #1: A thorough large scale study well conceived and produced rapidly in order to provide valuable insights to these personal responses to lockdown and Covid 19. The findings offer important practical guidance to planners that can help support coping strategies within communities and redressing inequity between communities more effectively.

Line 345 Is 'phenomenon' the right word? Would 'result' or something similar be better to avoid methodological confusion?

Reviewer #2: General Comments

This is a timely, well-crafted manuscript that highlights a well-designed study concerning CES and COVID-19. Most of my comments represent minor revisions. The one exception being the need to reconsider how those values that did not fall into either factor are used in analysis. I feel this requires a revised analysis. If this is not undertaken and significant citation, rationale, and discussion is provided instead, it should be listed as a limitation, at a minimum.

Overall, however, I feel this provides valuable insight and clearly represents a commendable, rigorous study.

Introduction

Overall comment. Quite a bit of recreation research related to activity and COVID-19 has been conducted to date. Some of this is cited, but additional discussion of this research base is merited in the introduction.

Examples.

Venter, Z. S., Barton, D. N., Gundersen, V., & Figari, H. (2020). Urban nature in a time of crisis: recreational use of green space increases during the COVID-19 outbreak in Oslo , Norway. https://doi.org/https://doi.org/10.31235/osf.io/kbdum

Derks, J., Giessen, L., & Winkel, G. (2020). COVID-19-induced visitor boom reveals the importance of forests as critical infrastructure. Forest Policy and Economics, 118, 102253.

O’Connell, T. S., Howard, R. A., & Hutson, G. (2020). The Impact of COVID-19 on Outdoor Recreation Participation in Canada.

Line 40. You may want to look to the work done following Hurricane Katrina for some insight in this area. It could be argued that we do know quite about how these large shocks impact the benefits we gain, values we place on nature, or perceive to gain from the natural world. You cite Rojas et al. (2017), but there is a larger body of work out there that directly speaks to this area you claim to be lacking.

Example:

Rung, A. L., Broyles, S. T., Mowen, A. J., Gustat, J., & Sothern, M. S. (2011). Escaping to and being active in neighbourhood parks: Park use in a post-disaster setting. Disasters, 35(2), 383–403. https://doi.org/10.1111/j.1467-7717.2010.01217.x

Lines 47-63. Given the scope of this paper, the authors should be more specific in their discussion of ecosystem services, namely, specifying that many of these values/benefits are recreational ecosystem services, a sub-class of CES. Rice et al. (2020) discuss the need for conceptualizing and measuring non-material benefits of recreation at length in their discussion of recreational ES. This distinction is needed as RES are unique in the way they are attained, through direct, active interaction with the natural world. Therefore, they can actually be more easily measured than some other forms of CES which can provision their benefits more passively. Additionally, as noted by Rice et al., study of RES is also important as it brings intangible recreation benefits into a common lexicon of ES—thus legitimizing the importance of these benefits to decision-makers.

Rice, W. L., Newman, P., Taff, B. D., Zipp, K. Y., & Miller, Z. D. (2020). Beyond benefits: Towards a recreational ecosystem services interpretive framework. Landscape Research. https://doi.org/10.1080/01426397.2020.1777956

Line 81. For COVID-19 specific findings concerning urban-rural gradients see:

Rice, W. L., Mateer, T. J., Reigner, N., Newman, P., Lawhon, B., & Taff, B. D. (2020). Changes in recreational behaviors of outdoor enthusiasts during the COVID-19 pandemic: Analysis across urban and rural communities. Journal of Urban Ecology, 6(1), 1–7. https://doi.org/10.1093/jue/juaa020

Materials & Methods

Line 107. How many other listservs and social media platforms were used in the data collection?

Line 113. What organization?

Lines 110-115. I would strongly encourage the authors to move this paragraph to the results section.

Line 120. How were these 15 activities selected? From previous research? Or the OIA or other list?

Lines 164-166. A citation is needed here.

Lines 188-189. Citation needed.

Line 202. A little confused here. Define “any combination”. Does this mean that if a participant designated one of the Factor 1 values as more important, they would be coded as 1 in the factor 1 dependent variable?

Line 202. I am similarly confused as to why values that were not included in either factor are included in both analyses: “we classified participants who ranked any combination of factor 1 values and values unassigned to either factor as more important during the COVID-19 restrictions as “1””. Language at the start of the paragraph does not suggest this. Additionally, it is unclear to the reader how many of those coded as “1” in the factor 2 dependent variable did not actually rank factor 2 values in their top three, but instead ranked unassigned values in their top three. Looking at Figure 2, the latter groups seems larger than the former. This has serious implications in your discussion of these results, and causes question towards the validity of this study. I understand that these unassigned values “loaded strongly with both factors,” but their inclusion in BOTH factors is not warranted and seems to undermine the validity of the study.

Results

Great job with the results section! I really like how the tables are displayed.

Line 238. These Pseudo R-squares are markedly low. I would recommend 1) discussing this statistic in the methods section (consider looking at Sapra (2014)), 2) being forthright about the low R2 in the results section, and 3) increasing the discussion of the limitations of this statistic and how these low R2s might impact the gravity of your findings in the limitations section.

Sapra, R. L. (2014). Using R2 with caution. Current Medicine Research and Practice, 4(3), 130–134. https://doi.org/10.1016/j.cmrp.2014.06.002

Discussion

General comment: Consider adding subheaders in this section. The content and organization of the discussion section is really well-done, but subheaders would aid the reader.

Line. 421. This is an excellent point! Really insightful and important. However, this paragraph is a little confusing to read. Consider expanding this paragraph to allow of more clarification of your points. Just want to make sure this finding and related discussion comes through clearly.

Line 458. Can you provide an alternative estimate in your results section? More discussion is needed here, as mentioned above.

Line 460. So should it be stated that these results should not be generalized outside of Vermont (or the region)?

Line 462. What common biases arise from convenience sampling? Discuss and cite.

6. PLOS authors have the option to publish the peer review history of their article (what does this mean?). If published, this will include your full peer review and any attached files.

Reviewer #1: **Yes: **Chris Loynes

Reviewer #2: No

---

## [Author Response · Author response to Decision Letter 0]

7 Nov 2020

Our full responses to each editor, reviewer, and journal comment are compiled in the Reviewer Response letter and Table included with this resubmission. We copy our summary from those documents of the major changes we have made in the following paragraphs. These fall into two main areas: 1) revised statistical analysis; and 2) expanded discussion of theory and policy implications.

Following on Reviewer 2’s urging to consider measures of model goodness of fit beyond the Pseudo R-square statistic, we present an updated and more conservative statistical analysis. We now limit our interpretation of six of our seven models to the largest levels of the dependent variables only (per Petrucci 2009), and engage with the rare events in logistic regression literature (Firth 1993, King and Langche 2001, Kosmidis and Firth 2011, and others). We discuss the rationale for this decision and provide supporting citations in both Methods and Discussion (Limitations) sections. We also provide a comment on the equity and policy implications of taking this conservative approach.

Following on your own Editor comments, we have developed our treatment of the theory implications of this work. In the Introduction, we better integrate our research into the literature on human-nature relationships and disasters, drawing on Rung et al. 2011, Derks et al. 2020, and others. We also situate our approach within the emerging literature on relational values (e.g. Klain et al. 2017) and comment on the potential to integrate this literature with value-change research. We also elaborate on our work in the context of Fish et al. 2016’s framework relating cultural practices in nature to values from nature. In the Discussion, we more thoroughly discuss the policy implications of this work in an equity context. We investigate the rare events logistic regression literature noted above in terms of our study’s relevance and shortcomings for informing equitable natural resource management policy. We also draw briefly on the policy sciences literature to support our observation of diverse values from shared experiences of natural resources (Clark and Vernon 2016, Morse and Clark 2019).

Additionally, we have made a number of smaller changes for clarity and to ensure that our formatting is consistent with PLOS ONE’s requirements. We are eager for your, and other editorial team members’, impressions of this revised manuscript, and much appreciative of your time and consideration.

---

## [Decision Letter · Decision Letter 1]

23 Nov 2020

PONE-D-20-23932R1

COVID-19 and human-nature relationships: Vermonters’ activities in nature and associated nonmaterial values during the pandemic

PLOS ONE

Dear Dr. Morse,

Thank you for submitting your manuscript to PLOS ONE. After careful consideration, we feel that it has merit but does not fully meet PLOS ONE’s publication criteria as it currently stands. Therefore, we invite you to submit a revised version of the manuscript that addresses the points raised during the review process.

Very satisfied with the way teh comments have been addressed. Some very minor final adjustments and it will be ready to accept for publication. 

We look forward to receiving your revised manuscript.

Kind regards,

Julia Martin-Ortega

Academic Editor

PLOS ONE

Reviewers' comments:

Reviewer's Responses to Questions

**Comments to the Author**

1. If the authors have adequately addressed your comments raised in a previous round of review and you feel that this manuscript is now acceptable for publication, you may indicate that here to bypass the “Comments to the Author” section, enter your conflict of interest statement in the “Confidential to Editor” section, and submit your "Accept" recommendation.

Reviewer #1: All comments have been addressed

Reviewer #2: (No Response)

2. Is the manuscript technically sound, and do the data support the conclusions?

Reviewer #1: Yes

Reviewer #2: Partly

3. Has the statistical analysis been performed appropriately and rigorously? 

Reviewer #1: I Don't Know

Reviewer #2: Yes

4. Have the authors made all data underlying the findings in their manuscript fully available?

Reviewer #1: Yes

Reviewer #2: Yes

5. Is the manuscript presented in an intelligible fashion and written in standard English?

Reviewer #1: Yes

Reviewer #2: Yes

6. Review Comments to the Author

Reviewer #1: Whilst my experience does not allow me to comment on the statistical methods, the discussion and recommendations have been significantly enhanced to reflect the value of the results which I have taken a face value. Additionally, the work is more strongly rooted in a range of appropriate theory for such a transdisciplinary study.

Reviewer #2: These revisions represent a much improved manuscript. I commend your willingness to revise your work, especially your analysis. The paper now reads much cleaner and I feel your conclusions are far more sound. Thank you for clarifying your treatment of items unassigned to either factor.

I have one last suggested revision, concerning statements made about the generalizability of your results. You say that "[O]ur results are likely indicative of relatively rural, White areas of the country." However, it is quite apparent that your sample is not representative of the median income-level across rural America-or of even America at large (https://www.census.gov/newsroom/blogs/random-samplings/2016/12/a_comparison_of_rura.html#:~:text=According%20to%20the%202015%20American,below%20the%20official%20poverty%20thresholds.), and even appears slightly higher than that of Vermont (https://www.census.gov/quickfacts/VT). As per my comment on the original submission, I feel this language concerning the representativeness of your sample needs further qualification. Your sample is significantly wealthier than rural America at large. Additionally, you collected information on your samples' political leanings, but did not include this information in your supplemental data file or in your discussion of your sample. With the polarization occurring around the pandemic and our related human behavior, this seems pertinent. I recommend that the language concerning the generalizability of your findings be limited to above average income households in Vermont and the surrounding region.

7. PLOS authors have the option to publish the peer review history of their article (what does this mean?). If published, this will include your full peer review and any attached files.

Reviewer #1: **Yes: **Dr Chris Loynes

Reviewer #2: No

---

## [Author Response · Author response to Decision Letter 1]

23 Nov 2020

We have addressed all reviewer and journal comments; please see the Response to Reviewers letter and table for a detailed treatment of the changes in substance and formatting that we have made.

---

## [Editor Report · Decision Letter 2]

26 Nov 2020

COVID-19 and human-nature relationships: Vermonters’ activities in nature and associated nonmaterial values during the pandemic

PONE-D-20-23932R2

Dear Dr. Morse,

We’re pleased to inform you that your manuscript has been judged scientifically suitable for publication and will be formally accepted for publication once it meets all outstanding technical requirements.

Kind regards,

Julia Martin-Ortega

Academic Editor

PLOS ONE
---

## [Editor Report · Acceptance letter]

4 Dec 2020

PONE-D-20-23932R2 

COVID-19 and Human-Nature Relationships: Vermonters’ Activities in Nature and Associated Nonmaterial Values during the Pandemic 

Dear Dr. Morse:

I'm pleased to inform you that your manuscript has been deemed suitable for publication in PLOS ONE. Congratulations! Your manuscript is now with our production department. 

Kind regards, 

on behalf of

Professor Julia Martin-Ortega 

Academic Editor

PLOS ONE